# Ultra-Stable Molecular Sensors by Sub-Micron Referencing and Why They Should Be Interrogated by Optical Diffraction—Part I. The Concept of a Spatial Affinity Lock-in Amplifier

**DOI:** 10.3390/s21020469

**Published:** 2021-01-11

**Authors:** Andreas Frutiger, Christof Fattinger, János Vörös

**Affiliations:** 1Laboratory of Biosensors and Bioelectronics, Institute of Biomedical Engineering, University and ETH Zürich, 8092 Zürich, Switzerland; afrutig89@gmail.com; 2Roche Pharma Research and Early Development, Roche Innovation Center Basel, 4070 Basel, Switzerland

**Keywords:** molecular sensors, chemosensors, biosensors, lock-in amplifiers, noise rejection, noise analysis, optical diffraction, focal molography

## Abstract

Label-free optical biosensors, such as surface plasmon resonance, are sensitive and well-established for the characterization of molecular interactions. Yet, these sensors require stabilization and constant conditions even with the use of reference channels. In this paper, we use tools from signal processing to show why these sensors are so cross-sensitive and how to overcome their drawbacks. In particular, we conceptualize the spatial affinity lock-in as a universal design principle for sensitive molecular sensors even in the complete absence of stabilization. The spatial affinity lock-in is analogous to the well-established time-domain lock-in. Instead of a time-domain signal, it modulates the binding signal at a high spatial frequency to separate it from the low spatial frequency environmental noise in Fourier space. In addition, direct sampling of the locked-in sensor’s response in Fourier space enabled by diffraction has advantages over sampling in real space as done by surface plasmon resonance sensors using the distributed reference principle. This paper and part II hint at the potential of spatially locked-in diffractometric biosensors to surpass state-of-the-art temperature-stabilized refractometric biosensors. Even simple, miniaturized and non-stabilized sensors might achieve the performance of bulky lab instruments. This may enable new applications in label-free analysis of molecular binding and point-of-care diagnostics.

## 1. Introduction

Molecular sensors—i.e., sensors that detect and characterize the binding of (bio-)molecules—are one of the key pillars of (bio-)analytics. They are widely used in drug screening and molecular diagnostics [1,2]. Molecular sensors enable the selective identification of molecules based on their binding characteristics and by this offer a way to study molecular interactions. Molecular sensors rely on molecular recognition [3,4]. Thus, they operate based on the binding affinity between a molecule of interest and the affinity probe, e.g., an antibody [5]. The binding affinity between the molecule of interest and the affinity probe is stronger than the affinity between probe and background molecules [6]. The main challenge is to translate molecular binding into a quantity that can be measured physically and to maintain the selectivity of the molecular recognition in the transduction. A property that is shared by all biomolecules and that can be detected with high sensitivity is their relatively high refractive index compared to water [7]. Therefore label-free optical biosensors, such as surface plasmon resonance (SPR), are so widely spread for studying molecular interactions [8,9]. One can correlate the refractive index change inside a test volume to a change of the number of molecules [8]. Traditional molecular sensors, as many other sensor types, are therefore integrative sensors, i.e., their output is the integral of the property of interest over the entire sensing volume. However, correlation is not causation and since all biomolecules have a higher refractive index than water, the type of molecule that is responsible for the change is not known *per se* [10]. In practice, for simple samples and for detection at comparatively high concentrations, the selectivity imparted by the recognition element combined with some control of or correction for non-specific binding suffices to associate a changing signal with changing concentration of analyte with reasonable confidence. However, this is not readily possible when very low detection limits are required and in applications involving the detection of an analyte in a complex biological fluid. Such sample matrices exert large and unpredictable non-specific responses. If a complex fluid is in contact with the sensor, the molecules of interest usually represent only a small fraction of the total biomolecular mass on the sensor, despite the affinity difference between the molecule of interest and the background molecules in the sample [11,12]. This is because the concentration of background molecules is several orders of magnitude higher than the concentration of the molecule of interest [13,14]. In addition, in classical binding experiments without any background molecules, the refractive index in the test volume is significantly affected by temperature gradients and medium inhomogeneities [15]. These effects are especially detrimental, since the molecules only occupy a tiny fraction of a typical sensor volume (a monolayer of molecules is ∼5 nm, while the height of the sensing volume in evanescent field sensors ranges from 100 nm to 300 nm) [16]. Thus, most state-of-the-art optical biosensors that are used to quantify molecular recognition are extremely cross-sensitive and require controlled conditions for their operation, otherwise the signal is buried in drift and noise [17].

Environmental noise, such as temperature fluctuations, is distributed in the time domain according to 1/f, where *f* is the frequency corresponding to the Fourier component of the noise spectrum [18,19]. The solution to the problem of detecting a small signal buried in environmental noise has been known since the 1930s and is widespread for temporal signals [20,21,22,23]. Lock-in amplification works by modulating the signal with a carrier frequency and multiplying the output of the sensor with a reference wave of the same frequency (also known as homodyne/heterodyne detection). One but not all the benefits of this detection principle is that by the action of modulation the signal power is shifted away from direct current (DC) to the modulation frequency where environmental noise is much lower. High-frequency lock-in amplifiers have enabled measurements of signals invisible with a purely integrative detector [24,25,26,27,28]. Up to now the lock-in principle was primarily restricted to the time domain using the transmission of electrical and optical signals. These signals can be easily modulated at a high temporal frequency.

On the other hand, binding signals of molecules cannot be modulated in time that simply. Their on and off rates are usually fixed by their molecular properties and the individual binding and unbinding events are not in phase. Thus, while temporal lock-in amplification can improve some of the noise characteristics in the readout electronics of these sensors, it cannot improve the signal to environmental noise ratio [29]. Since gradients in space and time are connected via the advection-diffusion equation, one can expect that the environmental noise in space is similarly distributed to the 1/*f* noise in the time domain [30]. Thus, most of the noise is long ranged (low or zero spatial frequencies) (Figure 1a) [18]. Integrative sensors have a spatial sensor transfer function that is centered around zero frequency (Figure 1b). For this reason, the signal largely overlaps with the noise and constitutes the heart of the drift and cross-sensitivity problem of most traditional refractometric biosensors and integrative sensors in general.

A widely applied strategy to reduce the zero-frequency component of the sensor transfer function, is to employ a differential measurement [15,31]. This is commonly known as referencing or calibrating. It is accomplished by a second integrative sensor and an electrical or digital subtraction of the two sensor outputs. However, this simple form of referencing is very inefficient, since even when macroscopically referenced, the transfer function of the sensor is still close to the origin of k-space and transfers most of the noise (Figure 1c). We shall see in this paper that for the improvement of the signal to environmental noise ratio a different lock-in, a spatial lock-in, is required.

As a temporal lock-in employs knowledge about the time dependence of a signal to separate it from noise, the spatial affinity lock-in uses the signal’s spatial dependence for the same purpose. Only when the binding signal is modulated at a high spatial frequency it can be efficiently separated from the noise (Figure 1d). A spatial lock-in is not a new idea *per se*, it relies on concepts that have already been developed in Fourier optics, optical information processing and holography. Also, some concrete applications to directly measure spatial coherence of light sources were demonstrated. However, it was only in the last two decades that two label-free optical sensors schemes were introduced that use a spatially modulated affinity pattern to improve their performance. One of the techniques are distributed reference surface plasmon resonance imaging sensors [15] and the other are diffractometric biosensors. The two classes are different in that distributed reference sensors are a digital lock-in whereas diffractometric sensors are an analog lock-in, similar to readout schemes already known in the time domain [18]. Their first results were encouraging, in that the resolution of non-temperature-controlled sensors was similar to temperature-stabilized commercial biosensors [11,15]. In addition, it was demonstrated that the noise decreases linearly with the splitting factor (analogous to a smaller referencing period). This confirms the 1/ξ character of the dominant noise (ξ is the spatial frequency) [15]. However, high spatial lock-in frequencies were achieved only recently [11,32,33,34,35]. This is because important foundations of a high-frequency spatial affinity lock-in, namely sufficiently coherent and spectrally stable light sources (i.e., the laser) and micron scale affinity patterning techniques of non-fouling adlayers (pioneered by photo-lithographic production of DNA chips) were only perfected in recent decades [32]. Yet, micron scale affinity patterning is mandatory but not sufficient for a high-quality spatial affinity lock-in. It is of crucial importance that no optical modulation or worse an affinity modulation for unwanted molecules is introduced on the sensor by the fabrication process [11]. To obtain the optimal affinity lock-in, affinity matching of the binding pair towards background molecules is also a necessity [36,37].

With this type of sensors emerging, it is pivotal to have a solid conceptual understanding of the basic working principle of the spatial affinity lock-in and promising techniques to implement it. These constitute the two aims of this paper. To keep our discussion simple, we do not address the noise sources in the measurement chain, i.e., source noise, such as straylight or wavefront stability, in our discussion. These noise sources are briefly discussed in part II and in detail elsewhere [38,39]. We start by introducing the concept of the spatial lock-in principle from a signal processing perspective. We derive the spectral distribution of environmental noise as well as the signal of integrative and locked-in sensors and show how locked-in sensors achieve reduced overlap of signal and noise bandwidths. After highlighting the analogies of the spatial lock-in to its time-domain counterpart, we will discuss the performance and the implementation of different spatial lock-in arrangements. In particular, we will learn that analog lock-in amplifiers (diffractometric sensors) have numerous advantages over digital lock-in amplifiers (distributed referenced surface plasmon resonance sensors). This is because analog spatial lock-in amplifiers sample Fourier space directly whereas digital spatial lock-in amplifiers sample in real space. It is advantageous to sample Fourier space directly because in Fourier space the signal and noise are spatially separated whereas in real space they overlap. All in all, the insights from the theoretical concepts presented in this paper, together with the experimental evidence and comparison to other established optical integrative sensors provided in part II [38], hint at the potential of spatially locked-in diffractometric biosensors to surpass state-of-the-art temperature-stabilized refractometric biosensors. Even simple, small and non-stabilized sensor devices might achieve the performance of current bulky lab instruments, a feature that may enable new applications in label-free analysis of molecular binding, environmental monitoring and point-of-care diagnostics.

## 2. Results and Discussion

Before we start discussing temporal and spatial lock-in amplification we give a brief introduction into linear system theory. In general, the input $x_{\mathrm{in}}\left(\vec{r},t\right)$ to any sensor can be written as a superposition of the signal $x_{s}\left(\vec{r},t\right)$ and the noise $x_{n}\left(\vec{r},t\right)$, which are both functions of space and time
(1)$$x_{\mathrm{in}}\left(\vec{r},t\right)=x_{s}\left(\vec{r},t\right)+x_{n}\left(\vec{r},t\right),$$
where $\vec{r}=\left(x,y,z\right)$ is the position vector and *t* is the time. The output of a linear system is the convolution of the input with the impulse response of the sensor $h\left(\vec{r},t\right)$
(2)$$y_{\mathrm{out}}\left(\vec{r},t\right)=h\left(\vec{r},t\right)\;*\;x_{\mathrm{in}}\left(\vec{r},t\right).$$

For the discussion of linear systems, it is convenient to adopt a frequency domain description because the convolution operation reduces to simple multiplication. In the Fourier domain, the sensor action writes
(3)$$Y_{\mathrm{out}}\left(\vec{\xi },f\right)=H\left(\vec{\xi },f\right)X_{\mathrm{in}}\left(\vec{\xi },f\right).$$

Here $H\left(\vec{\xi },f\right)$ is the sensor transfer function, i.e., the Fourier transform of the impulse response, and $\vec{\xi }=\left(\xi_{x},\xi_{y},\xi_{z}\right)$ is the position vector in reciprocal space. (The difference between spatial frequency and the more commonly used angular spatial frequency (k-vector) is merely a factor 2π ($\vec{k}=2\pi \vec{\xi }$).) $X_{\mathrm{in}}\left(\vec{\xi },f\right)$ and $Y_{\mathrm{out}}\left(\vec{\xi },f\right)$ are the spectral distributions of the input and output, respectively. A sensor transfer function $H\left(\vec{\xi },f\right)$ connects the spectral distribution of the input $X\left(\vec{\xi },f\right)$ to the spectral distribution of the sensor output $Y\left(\vec{\xi },f\right)$ as long as the system is linear. It is a Fourier domain description of the sensor action for any input. In addition, it is of interest to know how the power of signal and noise are transmitted through the system. This will allow calculation of the signal-to-noise ratio at the output of the sensor. In the time and space domain the total power is given by the integral of the autocorrelation function $xx\left(\vec{r},t\right)$ of the sensor input over the entire time period and real space
(4)$$P=\int_{0}^{\infty }\int_{0}^{2\pi }\int_{0}^{\pi }\int_{0}^{\infty }xx\left(\vec{r},t\right)r^{2}\sin\left(\theta \right)dtd\theta d\varphi dr.$$

In the frequency domain the total power is the integral of the power spectral density over all frequencies (Parseval–Plancherel identity)
(5)$$P=\int_{0}^{\infty }\int_{0}^{\infty }\int_{0}^{2\pi }\int_{0}^{\pi }\mathrm{PSD}\left(\vec{\xi },f\right)\xi^{2}\sin\theta d\theta d\phi d\xi df,$$
where in both cases we have changed to spherical coordinates for the integration ($x=r\sin\left(\theta \right)\sin\left(\varphi \right)$, $y=r\sin\left(\theta \right)\cos\left(\varphi \right)$ and $z=r\cos\left(\theta \right)$, $\xi_{x}=\xi \sin\left(\theta \right)\sin\left(\phi \right)$, $\xi_{y}=\xi \sin\left(\theta \right)$$\cos\left(\phi \right)$ and $\xi_{z}=\xi \cos\left(\theta \right)$ respectively). The power transmission of a system is therefore described by its mapping of the power spectral density (PSD) from input to output. It can be shown that the transfer function of the PSD is the squared magnitude of the sensor’s transfer function [40]. Thus, the PSD of the output as a function of the PSD of the input reads
(6)$${\mathrm{PSD}}_{y_{\mathrm{out}}}\left(\vec{\xi },f\right)={\left|H\left(\vec{\xi },f\right)\right|}^{2}{\mathrm{PSD}}_{x_{\mathrm{in}}}\left(\vec{\xi },f\right).$$

The power spectral density is the Fourier transformation of the autocorrelation function. Under the assumption of independence of signal and noise, the PSD of the input is the sum of signal and noise power spectral densities, ${\mathrm{PSD}}_{x_{\mathrm{in}}}\left(\vec{\xi },f\right)={\mathrm{PSD}}_{x_{s}}\left(\vec{\xi },f\right)+{\mathrm{PSD}}_{x_{n}}\left(\vec{\xi },f\right)$ [40]. The signal-to-noise ratio of the sensor can then be expressed by the sensor transfer function as well as signal and noise PSD
(7)$$\mathrm{SNR}=\frac{P_{S}}{P_{N}}=\frac{\int_{0}^{\infty }\int_{0}^{\infty }\int_{0}^{2\pi }\int_{0}^{\pi }{\left|H\left(\vec{\xi },f\right)\right|}^{2}{\mathrm{PSD}}_{x_{s}}\left(\vec{\xi },f\right)\xi^{2}\sin\theta d\theta d\phi d\xi df}{\int_{0}^{\infty }\int_{0}^{\infty }\int_{0}^{2\pi }\int_{0}^{\pi }{\left|H\left(\vec{\xi },f\right)\right|}^{2}{\mathrm{PSD}}_{x_{n}}\left(\vec{\xi },f\right)\xi^{2}\sin\theta d\theta d\phi d\xi df}.$$

Heuristically, the signal-to-noise ratio is maximized by employing the following three synergistic measures. First, shift the signal to a region in four-dimensional reciprocal space, where the noise PSD is lowest. Second, concentrate the signal PSD into the smallest frequency bandwidth possible. Third, reduce the bandwidth of the transfer function of the sensor as far as possible and match it to the signal bandwidth (matched filter) [40]. A good lock-in amplifier achieves all these simultaneously.

We have already depicted the PSD of environmental noise together with the PSD of the signal of different sensors schematically in Figure 1. In the following, we derive an analytical form of the PSD of environmental noise as well as the PSD of the signal of traditional and locked-in sensors.

### 2.1. Power Spectral Density of Environmental Noise

Environmental noise in molecular sensors is primarily due to temperature gradients, yet also concentration gradients and non-specific binding will play a role [17,41]. Nevertheless, we restrict our treatment of environmental noise to temperature gradients and for simplicity neglect convection. The formalism for concentration gradients is exactly the same, since as the temperature equation, the diffusion equation is also a parabolic partial differential equation. For a detailed discussion on intrinsic and extrinsic noise sources, see chapter 7 in [39]. The temperature equation can be written as
(8)$$\frac{\partial T\left(\vec{r},t\right)}{\partial t}=-D_{T}\nabla^{2}T\left(\vec{r},t\right)+\frac{q(\vec{r},t)}{\rho c_{p}},$$
where $D_{T}$ is the thermal diffusivity, ρ is the density, $c_{p}$ is the specific heat capacity under constant pressure and $q(\vec{r},t)$ are the heat sources. The solutions (modes) of this equation have the following form: $T_{\xi }=T_{0}e^{i2\pi \left(\vec{\xi }\cdot \vec{r}-ft\right)}$. By inserting this ansatz into Equation (Equation 8) and solving for the homogeneous solution ($q(\vec{r},t)=0$), we obtain the dispersion relation $f=i\xi^{2}2\pi D_{T}$, with $\xi :=\left|\vec{\xi }\right|=\sqrt{\xi_{x}^{2}+\xi_{y}^{2}+\xi_{z}^{2}}$ [42]. Solving the inhomogeneous equation $q(\vec{r},t)\neq 0$ yields the PSD of the temperature fluctuations $\langle T_{\xi ,f}{T_{\xi ,f}}^{*}\rangle$ as a function of the PSD of the heat sources $\langle Q_{\xi ,f}{Q_{\xi ,f}}^{*}\rangle$ (see Chapter 13 in [43] for a similar procedure at the example of the Langevin equation)
(9)$${\mathrm{PSD}}_{x_{n}}\left(\xi ,f\right):=\langle T_{\xi ,f}{T_{\xi ,f}}^{*}\rangle =\frac{\langle Q_{\xi ,f}{Q_{\xi ,f}}^{*}\rangle }{\rho c_{p}}\frac{4\pi^{2}D_{T}\xi^{2}}{{\left(4\pi^{2}D_{T}\xi^{2}\right)}^{2}+{\left(2\pi f\right)}^{2}}.$$

To proceed, we need to assume the functional form of the PSD of the heat sources. For simplicity and generality, we assume that the heat source process is wide sense stationary with zero mean and the autocorrelation of the heat sources has a negative exponential form with a characteristic length $L_{c}$ and timescale τ: $qq\left(r,t\right)=q_{0}e^{-\frac{r}{L_{c}}}e^{-\frac{t}{\tau }}$, where *r* is the norm of the position vector and *t* is time. Under these assumptions the one-sided power spectral density of the heat sources reads
(10)$$\langle Q_{\xi ,f}{Q_{\xi ,f}}^{*}\rangle =16\pi q_{0}\frac{{L_{c}}^{3}\tau }{{\left(1+{\left(2\pi \xi L_{c}\right)}^{2}\right)}^{2}}\frac{1}{1+{\left(2\pi f\tau \right)}^{2}},$$
where we have used the results from [44] for the PSD in three dimensions. By inserting this expression into Equation (Equation 9) we finally arrive at the power spectral density for environmental temperature noise
(11)$${\mathrm{PSD}}_{x_{n}}\left(\xi ,f\right)=16\pi \frac{q_{0}}{\rho c_{p}}\frac{{L_{c}}^{3}\tau }{{\left(1+{\left(2\pi \xi L_{c}\right)}^{2}\right)}^{2}}\frac{1}{1+{\left(2\pi f\tau \right)}^{2}}\frac{4\pi^{2}D_{T}\xi^{2}}{{\left(4\pi^{2}D_{T}\xi^{2}\right)}^{2}+{\left(2\pi f\right)}^{2}}.$$

For our discussion only the functional form of the PSD matters and the exact magnitude of the heat sources is irrelevant. Therefore, we use the following proportional expression for the remainder of this paper
(12)$${\mathrm{PSD}}_{x_{n}}\left(\xi ,f\right)\propto \frac{1}{{\left(1+{\left(2\pi \xi L_{c}\right)}^{2}\right)}^{2}}\frac{1}{1+{\left(2\pi f\tau \right)}^{2}}\frac{\xi^{2}}{{\left(4\pi^{2}D_{T}\xi^{2}\right)}^{2}+{\left(2\pi f\right)}^{2}}.$$

However, we do need a rough estimate of the spatial and temporal correlation lengths/times of the heat sources. These length/timescales are important because the terms in Equation (Equation 12) are constant up to a frequency that is equal to the inverse of the correlation length/time and then start to roll-off with $1/f^{2}$ or $1/\xi^{2}$ respectively (Lorenzian power spectrum). In practice, one will not notice this sharp transition because there are multiple dominant correlation mechanism including a few with extremely long time and length scales (i.e., seasonal variations) [18]. For our discussions it is nevertheless sufficient to assume just one dominant correlation mechanism and this one can be motivated heuristically. The heat sources that influence our sensor will neither be kilometers apart nor closer together than a few millimeters. Thus, a reasonable assumption for $L_{c}$ for a molecular sensor is in the order of centimeters. The dominant temporal correlation is likely shorter than a day and longer than milliseconds, thus a reasonable assumption for τ would be in the hundreds to thousands of second range. Since these correlation lengths are larger than the sensor dimension, the noise is 1/ξ distributed for all length scales smaller than the sensor. The same holds true for the temporal scales.

Equation (Equation 12) is the combined temporal and spatial power spectrum of the heat sources. To obtain just the spatial power spectral density, one needs to integrate Equation (Equation 12) with respect to *f* over the entire frequency spectrum to obtain
(13)$${\mathrm{PSD}}_{x_{n}}\left(\xi \right)\propto \frac{1}{{\left(1+{\left(2\pi \xi L_{c}\right)}^{2}\right)}^{2}}\frac{1}{\left(1+{\left(2\pi \xi \sqrt{D_{T}\tau }\right)}^{2}\right)}.$$

Due to the assumption of wide sense stationarity and $D_{T}$ being time and space independent we were able to carry out the integration. Equation (Equation 13) has two terms corresponding to the two correlation mechanisms of the heat sources (one spatial and one temporal): The first term with $\xi L_{c}$ is the spatial correlation due to the inhomogeneous spatial distribution of the heat sources. Its presence in the PSD was expected. The second term with $D_{T}\xi^{2}\tau$ is a spatial correlation mechanism caused by the temporal correlation of the heat sources (on-off switching) [30]. Thus, a temporal correlation mechanism will show up in the spatial frequency spectrum because the two domains are linked via the diffusion equation.

The $1/\xi^{6}$ dependence in Equation (Equation 13) shows us that the environmental temperature noise is mainly situated at low spatial frequencies. Therefore, it is beneficial to construct molecular sensors that have most of their signal at high spatial frequencies, i.e., through spatial modulation of the binding signal.

### 2.2. Signal Power Spectral Density of Integrative and Locked-in Sensors

We will now derive the PSD of the signal of the input of integrative and locked-in sensors. Since the signal is deterministic, the PSD is the square of the Fourier transform magnitude of the signal input
(14)$${\mathrm{PSD}}_{x_{s}}\left(\vec{\xi },f\right)={\left|X_{s}\left(\vec{\xi },f\right)\right|}^{2}.$$

The signal of a molecular sensor can be described by the multiplication of an envelope/shape function $s\left(\vec{r},t\right)$ and a modulation function $f\left(\vec{r},t\right)$
(15)$$x_{s}=s\left(\vec{r},t\right)f\left(\vec{r},t\right).$$

The shape function constrains the sensors extent in time and space. It is, therefore, dependent on the measurement time and the size of the sensor (zero outside of it). The modulation is a periodic function of infinite extent that modulates the signal within the sensor volume/experiment duration at a preferentially high spatial and/or temporal frequency. The signal of a molecular sensor based on a label-free optical measurement is a spatial and temporal change of the refractive index induced by the binding of molecules to the immobilized recognition sites. The shape and modulation function therefore describe the distribution of the refractive index due to the analyte in space and time during an experiment.

The signal’s power spectrum is obtained by computing the squared Fourier magnitude of the product of envelope and modulation functions
(16)$${\mathrm{PSD}}_{x_{s}}\left(\vec{\xi },f\right)={\left|X_{s}\left(\vec{\xi },f\right)\right|}^{2}={\left|\int_{-\infty }^{\infty }\int_{-\infty }^{\infty }\int_{-\infty }^{\infty }\int_{-\infty }^{\infty }s\left(\vec{r},t\right)f\left(\vec{r},t\right)e^{-2\pi i\left(\left(\vec{\xi }\cdot \vec{r}\right)+ft\right)}dxdydzdt\right|}^{2},$$
which can also be conveniently written as the convolution of the Fourier transforms of envelope *S* and modulation function *F*
(17)$${\mathrm{PSD}}_{x_{s}}\left(\vec{\xi },f\right)={\left|S\left(\vec{\xi },f\right)\;*\;F\left(\vec{\xi },f\right)\right|}^{2}.$$

As mentioned in the introduction, refractometric biosensors are integrative sensor. This means that they simply integrate the refractive index change within the entire sensor volume without any possibility for referencing within the sensing volume. Examples of integrative sensors represent nearly all unreferenced or macroscopically referenced (signal and reference sensors are further apart than the sensor dimensions, maybe even in different flow channels) refractometric sensors such as surface plasmon resonance (SPR) or waveguide-based biosensors. In integrative sensors, the modulation function is unity, and the signals PSD is then equal to the square magnitude of the Fourier transform of the envelope function only
(18)$${\mathrm{PSD}}_{x_{s},\mathrm{int}}\left(\vec{\xi },f\right)={\left|S\left(\vec{\xi },f\right)\right|}^{2}.$$

In a locked-in sensor the signal is modulated with a spatial modulation period Λ and/or temporal period $f_{0}$. Yet, only spatial modulation is important for environmental temperature and non-specific binding noise rejection. For algebraic simplicity, we assume a sinusoidal spatial modulation along the vector ${\vec{\xi }}_{g}=\left(\frac{1}{\Lambda },0,0\right)$ of the form $f\left(\vec{r},t\right)=\sin\left(2\pi \vec{r}\cdot {\vec{\xi }}_{g}\right)$, with Fourier transform $F\left(\vec{\xi },f\right)=\frac{1}{2i}\left(\delta \left(\vec{\xi }-{\vec{\xi }}_{g}\right)+\delta \left(\vec{\xi }+{\vec{\xi }}_{g}\right)\right)$, where $\delta \left(\vec{\xi }\right)$ is the Dirac delta function. If one performs the convolution in Equation (Equation 17) it is obvious that the action of the modulation is merely a splitting and a shift of the original signal input to the spatial frequencies $\xi_{x}=\pm \frac{1}{\Lambda }$ while its envelope and bandwidth remains unchanged. Mathematically, the PSD of the signal reads
(19)$${\mathrm{PSD}}_{x_{s},\mathrm{lock}\text{-}\mathrm{in}}\left(\vec{\xi },f\right)=\frac{1}{2}{\left|S\left(\vec{\xi }-{\vec{\xi }}_{g},f\right)\right|}^{2}+\frac{1}{2}{\left|S\left(\vec{\xi }+{\vec{\xi }}_{g},f\right)\right|}^{2}.$$

The smaller the modulation period the further the signal power is shifted away from zero frequency. Since the environmental noise decreases with increasing spatial frequency the signal-to-noise ratio is higher at high spatial frequencies, i.e., for small modulation periods. More figuratively speaking, interdigitated signal and reference regions should be together as closely and as numerous as possible within a given sensor.

### 2.3. Ingredients and Analogies of Temporal and Spatial Lock-In Amplifiers

We have seen in the last two sections that modulation shifts the PSD of the signal to the modulation frequency while preserving its bandwidth. Environmental noise is generally lower at these higher frequencies (Figure 1). However, this alone does not yet give a better signal-to-noise ratio of the molecular sensor. One effectively needs to construct a sensor transfer function such that it overlaps precisely with the signal bandwidth. In general, a lock-in can therefore be broken down into two essential operations. First, modulation shifts the signal power away from zero frequency to a frequency where noise is lower. In a second step, a bandpass filter that ideally matches the signal bandwidth is applied. The remaining Fourier components are then integrated both in the frequency domain and the space domain. The exact way how the bandpass filtering is accomplished depends on the implementation of the lock-in amplifier. There is, however, one essential difference between a lock-in and a traditional bandpass filter. In its essence, a lock-in is a tracking bandpass filter. It uses a reference wave to track the frequency (and depending on the implementation also the phase) of a signal of interest.

In the time domain this is achieved in an elegant way. By mixing (multiplying) signal and reference wave, the two signal sidelobes as well as the noise are shifted and one of the signal lobes is locked precisely to DC (Figure 2a,b) (The reference and the signal are usually taken from the same source and therefore have no mutual drift). On the other hand, due to the mixing, the noise has been shifted to the mixing frequency (Figure 2b). The second step is to use a filter to match the sensor transfer function to the signal bandwidths. Because half of the signal is located at DC, one can employ a very narrow bandwidth low-pass filter. The bandwidth is narrower than any bandpass filter that can be implemented by active RC (or LC) circuitry, where frequency stability problems limit the bandwidth [45]. In the time domain, the main motivation of the mixing with low-pass filtering over simple bandpass filtering is that one can construct a sensor transfer function that matches more precisely the signal bandwidths (at the cost of discarding half of the signal power).

In the spatial domain the basic ingredient for the most common implementation of a lock-in (analog lock-in) is the diffraction of a reference wave at ordered matter. Therefore, any kind of propagating wave phenomenon can be used for the implementation of a lock-in, e.g., electromagnetic or acoustic waves. A wave has a certain momentum ${\vec{\beta }}_{\mathrm{in}}$ that, if it is mixed with the grating/crystal momentum of ordered matter, can achieve a lock-in in reciprocal space (Figure 2c). Conservation of momentum manifests in the diffraction condition and leads to two diffracted waves that have a well-defined propagation direction relative to the one of the reference wave for every possible grating momentum vector ${\vec{\xi }}_{g}$ (Figure 2d). The selection of one of the two diffracted waves with propagation vector ${\vec{\beta }}_{\mathrm{out}}$ is mediated by a spatial filter at a Fourier plane (in the far field) that acts as a bandpass filter. The noise is also filtered out because its lower spatial frequencies primarily diffract in the forward direction (Figure 2d) [11]. In contrast to the temporal lock-in, the spatial lock-in cannot be locked to DC, because it is a vector lock-in and only the direction but not the magnitude of the reference wave momentum is changed. The direction is fixed by the diffraction condition. However, it is difficult to hit a fixed pinhole in space, especially since thermal and mechanical effects might change the diffraction condition slightly. Therefore, the spatial filter needs to track the diffracted signal. Although the pinhole could be moved in the Fourier plane, it is much easier to use an array detector and to track the signal by image registration. The spatial filter can then be applied digitally [11]. The tracking/registration operation therefore has the same effect as the DC lock of the signal in the time domain. More fundamentally, the lock-in in time is a one-dimensional energy lock-in whereas in space it is a three-dimensional momentum lock-in. This has some implications on the performance of spatial lock-ins of different dimensionality.

### 2.4. Performance of Spatial Lock-Ins of Different Dimensionality

To continue our discussion on the performance of spatial lock-ins of different dimensionality it is helpful to consider a concrete transfer function. For simplicity, we assume that the filter function perfectly overlaps with one of the lobes of the PSD of the signal and its squared Fourier magnitude has the same functional form. The bandwidth will therefore be determined by the envelope function of the sensor only. The envelope function is unity inside the sensor volume and the time span of interest:(20)$$s\left(\vec{r},t\right)=\begin{array}{cc} 1, & \mathrm{inside}\;\mathrm{volume}/\mathrm{time}\;\mathrm{of}\;\mathrm{interest}\\ 0, & \mathrm{outside}\;\mathrm{volume}/\mathrm{time}\;\mathrm{of}\;\mathrm{interest} \end{array}$$

We assume a rectangular truncated sensor with spatial dimensions $L_{x},L_{y},L_{z}$ and the sensor output being suddenly switched on and being switched off again after a time period $\tau_{\exp}$
(21)$$s\left(\vec{r},t\right)=\mathrm{rect}\left(\frac{x}{L_{x}}\right)\mathrm{rect}\left(\frac{y}{L_{y}}\right)\mathrm{rect}\left(\frac{z}{L_{z}}\right)\mathrm{rect}\left(\frac{t}{\tau_{\exp}}\right).$$

Here we have used the definition of the rect function as described in Appendix A. We have chosen the time axis such that the middle of the experiment corresponds to t=0. The transfer function of such an integrative sensor is therefore a product of four sinc functions centered at the origin along every coordinate
(22)$$H_{\mathrm{int}}\left(\vec{\xi },f\right)=L_{x}L_{y}L_{z}\tau_{\exp}\mathrm{sinc}\left(L_{x}\xi_{x}\right)\mathrm{sinc}\left(L_{y}\xi_{y}\right)\mathrm{sinc}\left(L_{z}\xi_{z}\right)\mathrm{sinc}\left(\tau_{\exp}f\right).$$

(In the case that the sensor was circular, $L_{x}L_{y}\mathrm{sinc}\left(L_{x}\xi_{x}\right)\mathrm{sinc}\left(L_{y}\xi_{y}\right)$ would need to be replaced by $\frac{D}{2}\frac{J_{1}\left(D\pi \xi_{r}\right)}{\xi_{r}}$, where $J_{1}$ is the Bessel function of the first kind order 1, *D* the diameter and $\xi_{r}=\sqrt{\xi_{x}^{2}+\xi_{y}^{2}}$.) The transfer function of a spatially locked-in sensor locked to the spatial frequency ${\vec{\xi }}_{g}=\left(\frac{1}{\Lambda },0,0\right)$ is simply the shifted version of Equation (Equation 22)
(23)$$H_{\mathrm{lock}\text{-}\mathrm{in}}\left(\vec{\xi },f\right)=L_{x}L_{y}L_{z}\tau_{\exp}\mathrm{sinc}\left(L_{x}\left(\xi_{x}-\frac{1}{\Lambda }\right)\right)\mathrm{sinc}\left(L_{y}\xi_{y}\right)\mathrm{sinc}\left(L_{z}\xi_{z}\right)\mathrm{sinc}\left(\tau_{\exp}f\right).$$

It is important to stress again that the transfer function is merely shifted by the lock-in and its bandwidth remains unaffected. Indirectly, the noise rejection depends on the ratio of the referencing length scale to the sensor dimension. Sub-micron referencing allows for roughly 1000 oscillations of the sine before its truncation. Therefore, its satellite peaks are well separated in k-space (Figure 1d) and resemble delta-distributions. Yet, if the sine is truncated after one oscillation only (a classical macroscopically referenced sensor) the sinc functions are very broad compared to their separation distance and even overlap (Figure 1c). Substantially more noise is picked up. In the time domain, this property is usually quantified as the quality factor *Q* as it measures the shift in frequency space relative to the FWHM around the center frequency and reflects the noise rejection capabilities of the system. One should also notice that the argument of the sinc function depends on the sensor/experiment extent along the corresponding direction and hence the full width half maximum (FWHM) of the power transfer function ${\left|H\left(\vec{\xi },f\right)\right|}^{2}$ along one dimension is $\mathrm{FWHM}=\frac{2.78}{L_{i}}$ with $L_{i}=\left\{L_{x},L_{y},L_{z},\tau_{\exp}\right\}$. Therefore, the concentration of the signal in reciprocal space and hence the noise rejection capabilities increase with experiment duration and the volume of the sensor. However, we cannot really influence the temporal extent/frequency content of Equation (Equation 22), since this is determined by the temporal characteristics of the molecular interaction to be investigated. (If kinetics are not important and we just perform an endpoint measurement τ is essentially how long we can average.) For the following discussion, we only consider the spatial dependency of Equations (Equation 22) and (Equation 23).

The performance of the spatial lock-in depends on five design variables, the spatial dimensions $L_{x}$, $L_{y}$, $L_{z}$ as well as the norm and direction of $\vec{\xi }$. The norm and direction of $\vec{\xi }$ in the sensor volume are important for the suppression of environmental noise (a shift does not change the bandwidth). $L_{x}$, $L_{y}$, $L_{z}$ affect the volume in reciprocal space (i.e., the bandwidth) and therefore influence the contributions of both environmental and white noise sources on the output ($P_{N}\propto \frac{1}{L_{x}L_{y}L_{z}}$). The lock-in is always in three dimensions but the envelope might exhibit unequal lengths $L_{x},L_{y},L_{z}$. In general, the envelope function of a sensor could resemble a 1D (e.g., fiber-based or strip waveguide-based [46]), 2D (e.g., classical surface-based biosensors [47]) or 3D function (e.g., photonic crystal sensors [48]). The shape of the envelope function in reciprocal space also depends on its dimensionality in real space (Figure 3a). Ultimately, the signal-to-noise ratio improves inversely with the sensor volume in real space. In other words, a higher signal-to-noise ratio comes at the expense of spatial resolution of the binding event. The dependence on the sensor volume also implies that a one-dimensional sensor needs to be much longer than the edge length of a 3D sensor to have the same performance $L_{1D}=\frac{L_{3D}^{3}}{\Delta y\Delta z}$. To give some numbers, for typical penetration depths of optical evanescent waves (Δy, Δz = 100 nm) and an edge length of 10 m of the 3D sensor, the one-dimensional sensor needs to be 10 cm long. This assumes that the entire evanescent field volume of the 1D sensor contains binding sites, which implies that it is of crucial importance that in 1D and 2D sensors most of the evanescent field volume is patterned with binding sites. Most biosensors are surface-based and hence two-dimensional ($L_{x},L_{y}\gg L_{z}$). As long as the grating vector is in the $\xi_{x},\xi_{y}$ plane, all locked-in sensors have the same signal-to-noise ratio (Figure 3b). This is because the sensor transfer function is symmetrical in $\xi_{x},\xi_{y}$. However, if the grating vector points along the z-direction (orthogonal to the sensor surface) the environmental noise rejection is severely compromised. The sensor transfer function is elongated along $\xi_{z}$ to such an extent that it now largely overlaps with the environmental noise. More figuratively, if the grating vector is along z in a 2D sensor, it is essentially not referenced anymore. For a 1D sensor this is even more restricted with only 2 points in reciprocal space that display the same signal-to-noise ratio (Figure 3c) because the sensor transfer function is elongated in two reciprocal spatial directions. On the other hand, in 3D the sensor transfer function is perfectly symmetric and every point on a sphere with radius $\left|\vec{\xi }\right|=\frac{1}{\Lambda }$ has the same signal-to-noise ratio (Figure 3d). Since there are many possible independent reciprocal volumes on the surface of this sphere, 3D sensors can easily be multiplexed. In real space, this would correspond to multiple overlayed affinity gratings with different grating vectors each giving rise to its own distinct Bragg reflection. In a way this represents an artificial crystal that can detect binding, i.e., a coherent molecular system. However, this shall not signify that 3D sensors are superior to 2D or 1D sensors, since other factors such as ease of readout or accessibility of binding sites by diffusion also influence the choice of sensor design. For instance, a 3D sensor would need to be rotated relative to the incident beam to read out all the different Bragg reflections.

To summarize, the following rules of thumb to maximize the separation of signal and noise can be stated: First, use the highest lock-in frequency possible for most efficient environmental noise rejection. The ultimate limit is a lock-in period that is twice as long as the largest molecule size because the signal and reference regions cannot be smaller than the molecule, otherwise already one molecule would lead to crosstalk (signal contribution to both signal and reference region at the same time). Since most solvated biological macromolecules have dimensions of 5–10 nm, this limit will be around 10–20 nm. Second, the grating vector should ideally point along the longest spatial direction of the sensor. Third, the sensing volume should be as large as possible to maximally constrain the signal in reciprocal space.

### 2.5. Implementations and Readout Modes of Spatial Lock-In Amplifiers

As for time-domain lock-ins, there are also many possible implementations of spatial lock-ins. Nevertheless, important distinctions for the signal-to-noise ratio is the nature of the wave that is modulated by the ordered matter and whether the lock-in is analog or digital. Furthermore, it is also relevant whether the lock-in is just in spatial frequency or in frequency and phase.

Provided that the sensor can be patterned with a sufficiently high spatial frequency, ultimately, the spatial lock-in frequency is limited by the wavelength of the wave that is used to probe the ordered matter. The short wavelength and gentle interaction with molecular matter make electromagnetic waves at optical frequencies especially suited to build high spatial frequency lock-ins with reasonable diffraction angles. Even higher lock-in frequencies could be achieved with electromagnetic waves in the UV and X-ray range or electron beams, but these waves are too energetic and therefore damaging to macromolecules. Besides optics, acoustics is also widely applied in label-free biosensing (quartz crystal microbalance (QCM), surface acoustic wave devices) [49,50]. However, acoustic waves have roughly 3 orders of magnitude longer wavelength than electromagnetic waves at optical frequencies. In addition, they exhibit high damping in water [51]. In conclusion, electromagnetic waves at optical frequencies are the wave phenomena of choice to build spatial lock-ins for molecular sensing.

We now briefly discuss analog and digital lock-in amplifier implementations and then provide a summary of their advantages and disadvantages. Analog lock-in amplifiers use the coherence of a reference wave and the Fourier transformation properties of free space propagation or lenses to acquire an image of a Fourier plane. (Fourier planes are always curved in reciprocal space and constitute a subset of the surface of a sphere with radius $\frac{{\vec{\beta }}_{\mathrm{out}}}{2\pi }$. The origin is such that it overlaps with the start of the incident beam momentum vector whereas the end of the incident beam momentum vector is located at the origin of k-space (Appendix B and Figure A1). Every point in the Fourier plane contains information about all the binding sites in the sensor volume [52]. Thus, by observing it, the entire sensor volume is monitored at this specific spatial frequency [11,36,53]. (One could also think of an analog lock-in that does not use coherence for the reference subtraction but somehow manages to readout all interdigitated signal and reference regions collectively in one measurement each. The measurement noise of such an implementation would still be higher because two measurement instead of one need to be performed. (The measurement noise would be uncorrelated, and its variance would be $\sigma_{\mathrm{sig}}+\sigma_{\mathrm{ref}}$.) A convenient way to create a Fourier plane without any additional optical components is to order the molecules in the shape of a focusing hologram (diffractive lens) [11,36]. The relevant Fourier component is then spread over an Airy disk of diameter $d_{\mathrm{Airy}}=\frac{1.22\lambda }{\mathrm{NA}}$, where NA is the numerical aperture ($\mathrm{NA}=\frac{\mathrm{Dn}}{2F}$, *D* is the diameter of the lens/sensor, *n* the refractive index and *F* the focal distance) of the focusing diffractive sensor and λ the wavelength. The further away the Fourier plane from the sensor the lower are the resolution requirements of the acquired image to isolate the relevant Fourier component.

A digital or also non-coherent spatial lock-in is essentially the acquisition of a high-resolution image of the entire sensor volume. From this image one then performs a digital Fourier transform and applies a digital bandpass filter (as in the time domain [18]). In biosensors, the digital lock-in approach has been demonstrated with sensors based on labeled molecules (e.g., the signal is mediated by a fluorophore) and the authors have reported significantly improved detection limits with the digital lock-in compared to integrative sensors [54]. In addition, any surface plasmon resonance imaging sensor using distributed referencing is essentially a digitally locked-in sensor [15]. Other options for a digital spatial lock-in could include arrays of LSPR (localized surface plasmon resonance) sensors [55] or arrays of slot waveguides [56]. These would need to be functionalized in an alternating manner and being separated at sufficient mutual distance to avoid coupling (crosstalk) between the sensing elements. For 2D sensors, one could also think of acquiring a high spatial frequency image with scanning probe techniques. However, due to their operation principles they are not really suited for the monitoring of ensemble properties of molecular interactions.

In summary, analog and digital lock-ins differ in that an analog lock-in samples the Fourier space representation of the sensor whereas the digital lock-in samples the sensor in real space and performs the Fourier transformation digitally (Figure 4). In real space the signal as well as the noise are distributed over the entire volume of the sensor. In Fourier space, the signal is not only concentrated to a small region but in addition noise and signal are separated. Due to these significant differences, analog lock-in amplifiers (diffractometric biosensors) have numerous advantages over digital lock-in amplifiers (refractometric biosensors using distributed referencing):

First, a digital lock-in needs to sample the entire sensing volume with a high spatial frequency such that the Nyquist criterion for the signal is met. Only then there is no aliasing, and the entire signal power is collected. On the other hand, the analog lock-in simply needs to integrate a small volume in Fourier space, because the entire signal power is localized there. The rest of Fourier space does not need to be acquired. As mentioned above, by adjusting the focal distance, the resolution requirements for detecting the relevant Fourier component can be rendered arbitrarily low. Thus, analog lock-ins can construct a sensor transfer function at high spatial frequency without the need for sampling the spatial domain with high resolution.

To put this into perspective, let us consider a 2D sensor of dimensions 1 mm × 1 mm with a lock-in period of 200 nm. To resolve the spatial modulation in real space a resolution of at least 100 nm is required (due to measurement noise it is even a bit smaller). To image the entire sensor 10,000 × 10,000 data points need to be acquired in a digital lock-in vs. 1 data point in the analog lock-in. This fact has multiple implications. First, imaging at optical frequencies at 200 nm resolution is challenging even with the best microscopes. Therefore, the spatial frequency of the digital lock-in will be at least one order of magnitude lower than in the coherent (analog) readout. In addition, current label-free optical biosensor face other constraints than the diffraction limit. Even with commercial instruments, the splitting factor in SPR cannot be increased below a few tens of microns due to constraints of the physical transduction principle, namely diffraction blurring by imaging the surface and the surface plasmon propagation length [15,57,58]. In other words, the free space propagator that enables the Fourier transformation in the analog lock-in works against the performance of the digital lock-in. The second implication is acquisition speed. With the current standard scientific sensor format of 1–10 megapixel, one ultimately requires stitching and refocusing to scan large sensor areas and therefore the acquisition will be considerable slower. The third drawback is dynamic range (digital versions of temporal lock-ins have the same issue (Section 6.3 in [18])). This drawback arises because the noise is measured as well. To give an example, consider a 12-bit camera: To acquire the signal in real space and a noise 1000 times higher than the signal we will lose 10 bits to the noise and can only measure the signal with the remaining two bits. This only gives an extremely coarse measurement output. On the other hand, in analog lock-ins the noise is spatially separated from the detector and is not acquired. Almost all the bits available can be used to digitize the signal. Fourth, there will be a measurement error for every data point. Therefore, a digital lock-in will have much higher measurement noise if the same detector is used as in the analog case. In addition, there is a fundamental difference in measurement error tolerance between refractometric and diffractometric biosensors as shown in part II [38]. Fifth, the readout system of an analog lock-in is considerable simpler. No high-resolution optical setups are required. In the simplest case one only needs a stable laser, an aperture and an integrative detector.

It is obvious that all these advantages must come at a cost. Yet, as it turns out the cost is actually information that is not required in label-free biosensors. Specifically, the exact location of one single binding event. Label-free interaction studies usually aim at determining ensemble properties of the interaction between two ensembles of biomolecules (affinity, concentration etc.) [59]. Thus, they rely on the determination of an average receptor occupancy (what fraction of receptors has bound a target) on the sensor surface. Usually, receptor occupancy is low in diagnostic applications because the concentration is 2–3 orders of magnitude lower than the affinity (dissociation constant $K_{D}$) of the molecular interaction [36,59]. Therefore, biosensors must collectively monitor the average occupation of millions of receptors at the same time without requiring knowledge of the exact state of an individual receptor. Another small disadvantage is that depending on the arrangement, the analog lock-in might require tuning to fulfill the diffraction condition. Yet, whenever the diffracted signal is a free space mode and the sensor is 2D, a simple array detector with digital tracking will be sufficient. The diffraction condition is automatically met at a slightly different angle that still hits the detector. In conclusion, analog lock-in amplifiers are more promising than digital ones when it comes to ease of instrumentation and environmental and measurement noise susceptibility.

Lastly, we want to investigate the difference between frequency and phase lock-ins. This is important because spatial lock-in amplifiers based on optical diffraction are only frequency lock-ins. A narrow sensor transfer function assures that only the carrier frequency and the frequency components in its vicinity are transduced. For simplicity, we now assume that we have a perfect frequency lock and that only the carrier frequency is transduced. Furthermore, we write the Fourier component of the carrier frequency of the sensor output after filtering in phasor notation. The resultant phasor $A_{r}$ still has two components: a signal phasor $A_{\mathrm{sig}}$ and a remaining noise phasor $A_{N}$ that have an arbitrary phase relationship ϕ to each other.
(24)$$A_{r}=A_{\mathrm{sig}}+A_{N}e^{i\phi }$$

This is best visualized in the complex plane (Figure 5). Without loss of generality, we have assumed the signal phasor to be aligned with the real axis. The difference between a frequency and phase lock-in is that a frequency lock-in only measures the magnitude of the resultant whereas the phase lock-in measures both magnitude and phase. To achieve a phase measurement, an external reference signal is required. The reference signal must be in a well-defined phase relationship with the signal and not pass through the system. The action of mixing is essentially a projection of the resultant phasor along the reference signal direction. By shifting the reference signal by 90${}^{\circ }$ both the in-phase and quadrature components of the resultant can be measured. In general, the signal-to-noise ratio will depend on how the measurement is performed and the noise properties. Under the assumption that the noise is time dependent and that the signal-to-noise ratio is the ratio between the signal power $A_{\mathrm{sig}}^{2}$ divided by the mean noise power $A_{N}^{2}\cos^{2}\left(\phi \right)$, a phase sensitive lock-in amplifier has the following signal-to-noise ratio: $\mathrm{SNR}=\frac{A_{\mathrm{sig}}^{2}}{A_{N}^{2}\cos^{2}\left(\phi \right)}$. This scheme is not easily accomplished in the spatial domain. In a spatial lock-in amplifier the spatial filter restricts the number of phasors that can pass through the detector. However, at optical frequencies only detectors that measure the light intensity, which is proportional to the squared amplitude, are readily available. The resultant phasor is therefore self-mixed and only its squared magnitude can be measured. The magnitude of the phasor is: $\left|A_{\mathrm{sig}}+A_{N}\right|=\sqrt{\left|A_{\mathrm{sig}}^{2}+2A_{\mathrm{sig}}A_{N}\cos\left(\phi \right)+A_{N}^{2}\right|}$ From the second term it is obvious that interference between signal and noise is present. Thus, signal and noise are no longer statistically independent, and one cannot define a proper signal-to-noise ratio for the frequency lock-in. As an approximation one can take the ratio between original signal and noise power without the interference term $\mathrm{SNR}=\frac{A_{\mathrm{sig}}^{2}}{A_{N}^{2}}$. Thus, the performance of a phase lock-in is not much better than a frequency lock-in if the phase angle is arbitrary and the noise is time dependent. However, the phase lock-in has significant advantages when the noise phasor is mostly static and can be measured before the signal is applied. This is the case for the speckle background of diffractometric biosensors in real time measurements [11]. A detailed noise analysis would involve concepts from speckle physics; however, this is beyond the scope of this paper. Yet briefly, the speckle background due to surface roughness is the noise phasor. Since it arises from a random phasor sum its length is Rayleigh distributed and its phase is uniform over (−π,π) [38,52]. If the phase information is unknown, the resultant phasor (signal + noise) has a length that is Rician distributed. The statistics of the Rician distribution determines the uncertainty of the pure frequency lock-in. In the phase lock-in, the noise phasor length and phase uncertainty will only be limited by the precision of the readout instrumentation and the stability of the illumination. In part II, we discuss this point more in depth [38].

## 3. Conclusions

Environmental noise due to temperature, buffer changes and non-specific binding is the limiting noise source in integrative molecular sensors that measure molecular recognition through the detection of binding. Integrative molecular sensor designs are sub-optimal because their sensor transfer function is situated at the origin of reciprocal space where most of the power of environmental noise is situated. In this paper, we introduced the “spatial affinity lock-in amplifier” as an effective design principle to reject environmental noise. Sensors employing the spatial affinity lock-in principle show better noise performance than traditional sensors because they shift the signal power to higher spatial frequencies where the noise is lower. In general, the higher the spatial modulation frequency the better the signal to environmental noise ratio. High spatial modulation frequencies are achieved by analog lock-ins based on diffraction of electromagnetic waves at optical frequencies, i.e., diffractometric biosensors. The diffractometric readout principle has intrinsic advantages because the wave nature and coherence of light does all the necessary computation. Coherence performs the reference subtraction, because the phase naturally oscillates over one signal and reference region. It performs the Fourier transform via free space propagation such that global correlation properties, i.e., all signal and reference regions, can be observed conveniently in a single point of a Fourier plane of an optical system by one detector [60,61,62]. On the other hand, digital lock-ins based on distributed referenced refractometric sensing principles (e.g., surface plasmon resonance imaging) have lower lock-in frequencies, lower dynamic range, higher measurement errors and need to sample the entire sensor volume with high resolution [38]. Further improvements in the signal to environmental noise ratio of diffractometric spatially locked-in sensors might be possible by employing 3D sensor designs and developing a spatial phase lock-in. In summary, the results in this paper suggest that robust label-free sensors should employ a spatially locked-in design to enable clean transmission of chemical signals in noisy, biologically complex environments. This fact already enabled novel applications such as the label-free observation of specific GPCR activation in living cells, simply because the lock-in frequency is higher than the characteristic inverse length scale of the cell, i.e., its size [33,34]. Therefore, dominant sources of noise induced by the cell such as adhesion changes, cell shape changes, mass redistributions etc. are efficiently filtered out. Despite this, we believe that such novel drug screening applications are just the beginning and the potential application scope of spatially locked-in sensors is vast as further outlined in part II [38].

## Figures and Tables

**Figure 1 sensors-21-00469-f001:**
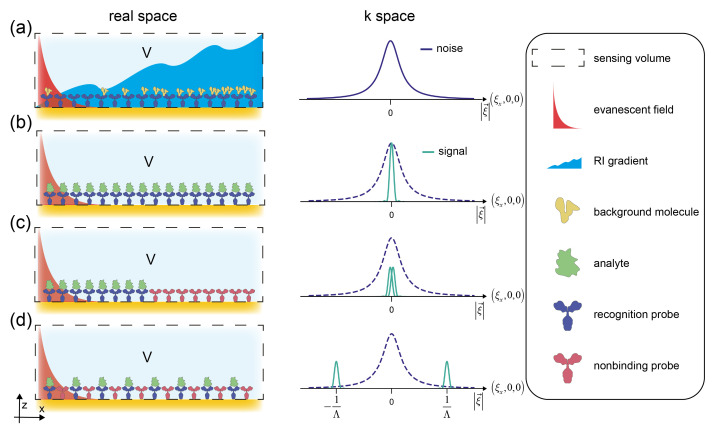
The spatial affinity lock-in concept. Environmental noise and signal distribution are shown for different sensor designs. In all cases, we assume a sensor that interrogates a volume close to its surface and detects changes in the refractive index within that volume. The second column shows the reciprocal space (also known as momentum or k-space) distribution of binding signal and environmental noise. (**a**) Typical environmental noise sources such as temperature gradients or non-specific binding of background molecules, are ‘long’ ranged. They usually exhibit a dominant correlation mechanism and display a Lorenzian shape in momentum space. (**b**) An integrative sensor with the analyte molecules bound uniformly has a spectral distribution of the signal centered around zero spatial frequency in reciprocal space where most of the environmental noise is situated. (**c**) A traditional referenced sensor still has most of the signal at DC and there is a significant spectral overlap between the two signal lobes because the modulation period is of similar size as the sensor dimension. (**d**) A locked-in sensor has a lock-in period that is much smaller than its lateral dimension. The signal lobes are clearly separated, and the signal is situated at spatial frequencies where there is hardly any environmental noise. As a remark, because we simply want to illustrate the concept, we have neglected a zero-frequency component that is because we cannot put a negative mass analyte molecule onto the reference. Also, we neglected all higher harmonics that are due to the rectangular distribution of biomolecular mass in this figure.

**Figure 2 sensors-21-00469-f002:**
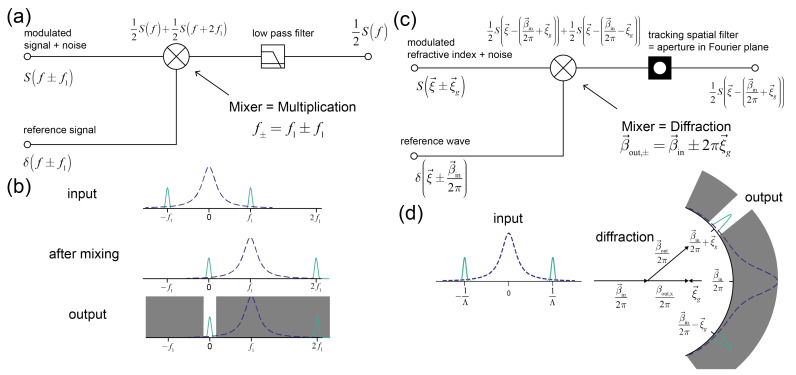
Analogies between a temporal (**a**,**b**) and a spatial lock-in (**c**,**d**) illustrated at simplified schematics a) The temporal lock-in uses a reference wave $\delta \left(f\pm f_{1}\right)$ from the same source as the signal (frequency $f_{1}$) that is mixed (multiplied) with the modulated and enveloped signal $S\left(f\pm f_{1}\right)$. This produces the sum and the difference of the two frequencies $f_{1}$ and locks one of the lobes of the signal to DC. A low-pass filter then isolates the DC component and produces an output that is proportional to the signal amplitude. (**b**) schematic depiction how the signal and the noise are shifted by the mixing and filtered by the low-pass filter. (**c**) In the spatial lock-in system a reference wave with momentum ${\vec{\beta }}_{\mathrm{in}}$ is mixed (diffracted) with the grating/crystal momentum of ordered matter (enveloped because of finite spatial extent $S\left(\vec{\xi }\pm {\vec{\xi }}_{g}\right)$) and produces two diffracted waves ${\vec{\beta }}_{\mathrm{out}}$. A spatial filter in a Fourier plane is then used to isolate one of these diffracted waves. (**d**) Schematic depiction how the modulated input and noise are mapped onto different angular directions by diffraction. The low frequency environmental noise is primarily scattered in the forward direction. A spatial filter isolates the signal component of interest.

**Figure 3 sensors-21-00469-f003:**
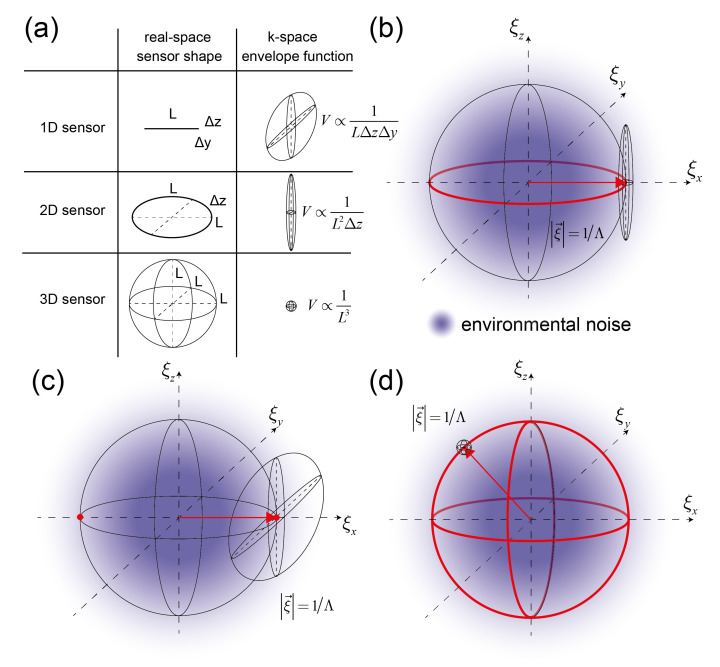
Transfer functions of spatial lock-ins and schematic 1/ξ (environmental) noise (**a**) Transfer functions of 1D, 2D and 3D sensors. The volume of the envelope function in reciprocal space decreases with increasing dimensionality of the sensor. For a given lock-in frequency, the amount of picked up noise decreases from 1D to 3D sensors. (**b**) 2D sensor transfer functions with equal signal-to-noise ratio and the same lock-in momentum magnitude $\left|\vec{\xi }\right|$ form a circle in the $\xi_{x}$, $\xi_{y}$ plane (**c**) 1D sensor transfer functions with equal SNR and the same lock-in momentum magnitude are just two points in reciprocal space. (**d**) 3D sensor transfer functions with the same SNR and same lock-in momentum represent an entire sphere in reciprocal space. This allows for many possible sensor realizations.

**Figure 4 sensors-21-00469-f004:**
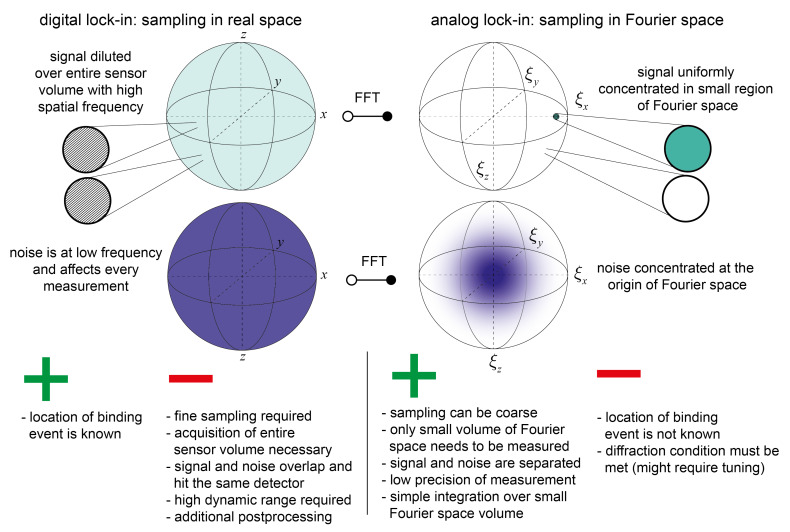
Advantages and disadvantages of digital and analog spatial lock-in amplifiers. Digital lock-in amplifiers such as surface plasmon resonance sensors using distributed referencing acquire the signal in real space whereas analog lock-in amplifiers (diffractometric sensor) sample directly reciprocal space. Due to this simple difference analog lock-in amplifiers have numerous advantages over digital lock-ins.

**Figure 5 sensors-21-00469-f005:**
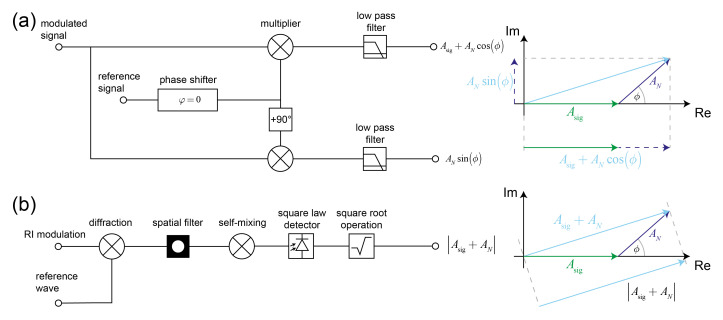
Difference between phase lock-in and pure frequency lock-in (**a**) A phase lock-in requires a reference signal with a fixed phase relationship to the signal. By mixing the signal with the reference and a 90${}^{\circ }$ shifted reference one can obtain the in-phase and quadrature components of the resulting phasor. This allows both magnitude and phase of the resultant to be determined. (**b**) A frequency lock-in such as a diffractometric biosensor loses the phase information because only the square magnitude of the resultant phasor can be detected (square law detection).

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
