# Peer review of "Ultra-Stable Molecular Sensors by Sub-Micron Referencing and Why They Should Be Interrogated by Optical Diffraction—Part I. The Concept of a Spatial Affinity Lock-in Amplifier"

_sensors, 2021, doi:10.3390/s21020469_

Round 1

Reviewer 1 Report

This article presents the potential of an original optical method for molecular sensors which does not involve the use of a reference nor a calibration for their stabilization. The biosensor transduction system described here is based on the spatial frequency modulation of the signal by a diffractive device. This work is based on Fourier theory to highlight the advantages of such a technique in terms of signal to noise ratio (SNR).

Some general comments:

The subject is well introduced and points out the problems due to the detection of a weak signal in a noisy medium (low concentration of the molecules of interest compared to that of the other molecules in the fluid, temperature gradients, inhomogeneity of the medium). Modulating and filtering unwanted frequencies is an elegant solution to bypass these problems. The spatial modulation method proposed here improves performance compared to temporal modulation which presents drawbacks (incoherent temporal signal between molecules, overlap of noise and signal spectra).

The method is clearly explained, demonstrating its potentiality. It is developed in 5 parts: 1. Power spectral density of noise, 2. Signal power spectral density (without or with spatial modulation), 3. Analogy between temporal and spatial modulators, 4. Performance of spatial modulators of different dimensions, 5. Implementation of spatial frequency modulators. The rules for a good performance of spatial frequency modulation are then summarized in three points: 1. Highest modulation frequency without exceeding the value which corresponds to twice the molecule size; 2. Modulation according to the largest dimension; 3. Probed volume as large as possible. All these points are clearly explained and the advantages of the proposed method are compared to those of other methods.

The conclusion is also well-argued and proves the ability of spatial frequency modulation to detect weak signals in noisy environments. In summary, this article sheds new light on spatial frequency modulation and demonstrates its interest for its use in bio-sensors. It is clearly written and well organized and deserves to be published. The revisions listed below are suggested to further improve its quality.

Others comments and questions:

Some information and numerical values, however, may be added such as:

  1. The impact on the SNR of the characteristics of the source, in particular the influence of its spectral width, polarization and stability, could be commented. In a sensor indeed, the entire measurement chain is important to qualify its efficiency to extract the signal.
  2. Apart from the thermal noise, what other phenomena could disturb the signal detection and in what proportion? In particular at line 235, the spatial modulation is assumed to be infinite and perfect while in practice the sine is truncated and can be distorted. The spatial limitation and the potential irregularities (roughness) of the grating can make the filtering difficult and contributes to SNR decrease.
  3. The measurement range, the repeatability and the resolution of the sensor are also important quantities. The performances of the spatial frequency modulation [1] could be discussed comparatively to those of others optical methods [2] like the interferometric [3, 4] or the resonant optical systems [5, 6] and the SPR method widely used in biosensor applications [7, 8].
  4. Could a simple device with simultaneous spatial and temporal modulations be designed to further improve the SNR?

Minor corrections:

Lines 35, 482; 123, 429, 487, 507, 513, 517: [3? ], [44? ]; [?]: ? corresponds to part 2

Lines 75, 99, 500: characteristics, digitial, substraction

Lines 111 (116); line 349: In the first (second) part: there is only one part: section 2; section 3 instead of sub-section 2.5?

Equation 10: check if it is not 16 instead of 32?

Equation 11: PSD is normalized here, not the same as in equation 10?

Lines 188 +189: C is missing in the equation and PSD = PSDxn inside integral

Lines 212, 213 and Equation 13: Can Equation 12 be integrated whereas the time and space variables are linked?

Line 340: reflections instead of reflexes?

Figure 3: to be included to the previous paragraph 2.4

Line 373: lens instead of sensor

Line 568: Reference 20.: To complete

[1]           K. Tsukamoto, N. Morinaga, Heterodyne Optical Detection / Spatial Tracking System Using Spatial Field Pattern Matching Between Signal and Local Lights, Electronics and Communications in Japan, Part 1, Vol. 77, No. 12, 1994

[2]           A. F. Fernandez Gavela, D. Grajales García, J. C. Ramirez, L. M. Lechuga, Last Advances in Silicon-Based Optical Biosensors, Sensors 2016, 16, 285; doi: 10.3390 / s16030285

[3]           G. H. Cross, A. A. Reeves, S. Brand, J. F. Popplewell, L. L. Peel, M. J. Swann, N. J. Freeman, A new quantitative optical biosensor for protein characterization, Biosensors and Bioelectronics 19 (2003) 383_/390

[4]           K.E. Zinoviev, A.B. Gonzalez-Guerrero, C. Dominguez, L.M. Lechuga, Integrated Bimodal Waveguide Interferometric Biosensor for Label-Free Analysis. Journal of Lightwave Technology, (2011). 29. 13: 1926; doi: 10.1109/JLT.2011.2150734

[5]           M. Kyoung Parka, J. Sheng Keea, J. Yiying Quaha, V. Nettob, J. Songa, Q. Fanga, E. Mouchel La Fosseb, G.-Qiang Lo, Label-free aptamer sensor based on silicon microring resonators, Sensors and Actuators B 176 (2013) 552– 559

[6]           D.-X. Xu, M. Vachon, A. Densmore, R. Ma, S. Janz, A. Delâge, J. Lapointe, P. Cheben, J. H. Schmid, E. Post, S. Messaoudène, J.M. Fédéli, Real-time cancellation of temperature induced resonance shifts in SOI wire waveguide ring resonator label-free biosensor arrays, October 2010 / Vol. 18, No. 22 / OPTICS EXPRESS 22867

[7]           M. Piliarik, J. Homola, Self-referencing SPR imaging for most demanding high-throughput screening applications, Sensors and Actuators B 134 (2008) 353–355, https://doi.org/10.1016/j.snb .2008.06.011

[8]           P. Hlubina, D. Ciprian, Spectral Phase Shift of Surface Plasmon Resonance in the Kretschmann Configuration: Theory and Experiment, Plasmonics (2017) 12: 1071–1078, DOI 10.1007 / s11468-016-0360-9

Author Response

See attached response letter.

Reviewer 2 Report

The manuscript seems overall fine, I suggest all authors to read it carefully. There seems to be some typos such as reference [3?], "were" should be probably "where". Also the word "thus" is used too often, "i.e.", starting sentences with prepositions, inserted sentences etc. The issues I found are marked with yellow color.

Apart of that I do not have any comments but I have to admit, I am not really an expert in the field, even we do use routinely lock-in amplification technique, but we do not develop those methods, we are "only" users.

Author Response

See attached response letter.

Reviewer 3 Report

This manuscript reports a theoretical and conceptual investigation of the use of a spatial lock-in amplifier approach with optical diffraction transduction to achieve improved sensor performance with the intention of using the devices as an optical biosensor transduction platform. The paper is reasonably clearly written and lays out the rationale for the approach and a consideration of the factors that are optimised to achieve the best transduction performance. The included figures provide a clear schematic representation of key concepts. The format advocated by the authors is likely to be of significant interest as a means of reducing the impact of noise on biosensor measurements and achieving robustness of signal output with a minimum of thermal or environmental control. There are some points for the authors to consider and these are outlined below.

  1. Page 2 lines 45-47 “However, correlation is . . . is not known [9].” Whilst fundamentally this is a reasonable point, in practice, analysts can associate changing signal with concentration of analyte with reasonable confidence unless the matrix non-specific binding is exceedingly variable or dominates responses even after attempts at correction. For simple samples and for detection at comparatively high concentrations, the selectivity imparted by the recognition element combined with some control of or correction for non-specific binding suffices. Where this really becomes an issue is where very low detection limits are required and where sample matrices exert large and unpredictable non-specific binding responses.
  2. Page 3 lines 107-108 “. . . and micron scale . . . the last decades.” This statement suggests that no matter the degree of sophistication in your optical format, to achieve the sensor performance improvements from this technique you still need a very fine degree of control over surface fabrication and/or functionalisation. In other words, this statement is suggesting that micron-scale resolution of sensor surface patterning and high-frequency spatial affinity lock-ins go hand-in-hand. How confident can you be that such surface fabrication is available at a quality standard such that it would not compromise the improvements in output signal quality and sensor performance posited in this paper? This issue is perhaps worthy of a bit more discussion.
  3. Page 3 line 123 “. . . [? ] . . .” This referencing needs to be tidied up. If the manuscript to which you refer has not been published yet then you should mark it as a currently ‘unpublished’ or ‘under consideration’ manuscript in the reference list.
  4. Page 16 lines 511-513 “This fact already enabled . . . its size [? ].” If you are quoting this example in the conclusions as work already done, then a definite proper literature reference should be provided.
  5. Figure A1. Two of the labels on this figure are too small to see clearly and need to be enlarged.

Author Response

See attached response letter.
